# Single-cell multiomics data analysis of potential receptors and therapeutic drugs for epilepsy patients comorbid with depression

**Guiqin Bai**[1☯], **Xuerong Zhou**[2☯], **Cheng Xiong**[2], **Xi Kang**[2], **Ruiqi Huang**[2], **Dazhang Bai**[2], **Peilin Zhao**[2], **Tao Peng**[1], **Cheer Muer**[1], **Guohui Jiang**[2], **Shushan Zhang**[2]*

1 Department of Basic Medicine and Forensic Medicine, North Sichuan Medical College, Nanchong, Sichuan, China, 2 Department of Neurology, Affiliated Hospital of North Sichuan Medical College, Institute of Neurological Diseases, North Sichuan Medical College, Nanchong, Sichuan, China

☯ These authors contributed equally to this work.
* susan448@163.com

## Abstract

Depression frequently cooccurs with epilepsy (EP) and has become a focus of clinical management, but effective pharmacological interventions remain limited. In this study, single-cell RNA sequencing (scRNA-seq) data were analyzed to identify changes in oligodendrocyte precursor cells (OPCs) in EP and major depressive disorder (MDD) patients, and intercellular communication and trajectory analyses were performed. Key therapeutic targets and pathways were identified via differentially expressed genes (DEGs), protein-protein interaction (PPI) networks, and gene ontology (GO) enrichment. A connectivity map (CMap) was generated to identify optimal drugs. Molecular dynamics simulation (MDs) and cell thermal stability migration assay (CETSA) were conducted to evaluate protein-drug interactions. The results revealed significant changes in gene expression in OPCs, with neuroligin3 (NLGN3)-neurexin (NRXN) signaling being the main pathway involved. Three hub genes correlated with NLGN3 were enriched in oxidative phosphorylation and mTORC1 signaling. Ziprasidone could effectively treat EP with MDD by strongly binding to NLGN3, forming two hydrogen bonds with a binding energy of −7.5 kcal/mol. This stable interaction was further confirmed by MDs and CETSA experiments. In conclusion, the NLGN3 protein interacts with ziprasidone to form stable complexes, which may activate the NLGN3-NRXN signaling pathway in OPCs and enhance synaptic remodeling by reducing neuroinflammatory responses.

## Introduction

Epilepsy (EP) is the most common neurological disease. At present, approximately 60 million people worldwide suffer from active epilepsy [1]. Depression is arguably the most common psychiatric comorbidity of epilepsy, especially temporal lobe epilepsy,

**Data availability statement:** Publicly available datasets were analyzed in this study. These data can be found in the GSE213982 (https://www.ncbi.nlm.nih.gov/geo/query/acc.cgi/acc=GSE213982), and GSE190452 (https://www.ncbi.nlm.nih.gov/geo/query/acc.cgi/acc=GSE190452) datasets.

**Funding:** This study was supported by the Doctoral Start Fund of North Sichuan Medical College (CBY22-QDA18), the Special Project for Basic Research in Traditional Chinese Medicine of Sichuan Province (25MSZX560), the Scientific Research Development Project of the Clinical Medical College Affiliated Hospital of North Sichuan Medical College (2023PTZK012), the Sichuan Science and Technology Program (2023NSFSC0709, 2024NFSC0490), the Guangdong Basic and Applied Basic Research Foundation (2023A1515110847), the Jiangmen Basic and Applied Basic Key Projects (2320002001026), and the Innovation and Entrepreneurship Training Program for College Students of North Sichuan Medical College (202510634011).

**Competing interests:** The authors have declared that no competing interests exist.

affecting approximately 30% of patients [2]. Moreover, in at least 30% of patients, EP cannot be controlled by currently available antiseizure medication [3]. Therefore, the clinical treatment of comorbid EP and major depressive disorder (MDD) has received increasing attention and has attracted widespread concern. The pathogenesis of comorbid EP and MDD involves multiple factors, such as neuroendocrine disorders, neuroinflammation, neurotransmitter abnormalities and synaptic plasticity. Neuroinflammatory factors such as TNF-α, IL-1β, and IL-6 aggravate damage to nerve cells by promoting inflammatory responses, further exacerbating epileptic seizures and de-pressive symptoms [4–6]. Additionally, nerve damage due to epileptic seizures can alter synaptic plasticity, with depression potentially worsening this effect, creating a vicious cycle [7]. However, owing to the relatively complex biological mechanism underlying the comorbidity of EP and MDD, the specific pathogenesis remains unclear, and no effective therapeutic drugs are currently available [3, 8]. Therefore, there is an urgent need for innovative treatment strategies aimed at increasing the quality of life and survival rates of individuals with EP and comorbid MDD.

Oligodendrocyte precursor cells (OPCs) play crucial roles in the central nervous system and can not only differentiate into oligodendrocytes to form the myelin sheath but also differentiate into astrocytes under specific conditions [9]. The deletion of Tcf4 and lineage-specific presenilin enhancer 2 can cause OPCs to differentiate into astrocytes, altering the glial cell distribution in the brain [10, 11]. Astrocytes affect the survival, proliferation and differentiation of OPCs by secreting soluble factors and promoting remyelination of axons [12, 13]. Recent research has indicated that astrocyte cannabinoid receptor 1 may reduce anxiety and depression-like behaviors in mice [14]. OPCs can directly establish synaptic connections with neurons, clear excess synapses through phagocytosis, and participate in synaptic pruning and neural circuit refinement together with microglia and astrocytes [15–17]. Recurrent seizures in patients with intractable EP and animal models can impair myelin regeneration in the hippocampus and cortex, along with the differentiation and maturation of OPCs [18, 19]. Moreover, impaired OPC development can cause demyelination, worsen nerve dam-age and inflammation, and trigger epileptic seizures [18, 20]. Thus, OPCs and astrocytes play key roles in the pathogenesis of EP and MDD.

NLGN3, a unique neuroligin (NLGN) subtype, is the ligand of neurexin (NRXN) [21]. It is highly expressed in neurons and OPCs and is located at the postsynaptic membrane of neurons [22–24]. Its structure comprises three main parts: the extra-cellular cholinesterase-like domain, the O-glycosylation site, and the transmembrane region [25]. As a synaptic adhesion molecule, it plays a crucial role in synapse formation and regulation in the central nervous system, with abnormal expression linked to neuro-logical disorders such as EP and MDD. NLGN3 influences excitatory synaptic trans-mission by clustering postsynaptic AMPA receptors with the PSD95 complex, promoting tumor growth and seizures [26]. Photo-biomodulation improves synaptic and cognitive function and ameliorates epileptic seizures by inhibiting the downregulation of NLGN3 expression [27]. In the chronic unpredictable mild stress model, NLGN3 knockout significantly reversed depressive-like behaviors, including weight loss, reduced sucrose intake, and decreased immobility in forced swimming

and tail suspension tests. Additionally, the knockout of NLGN3 was associated with inactivation of the NLRP3 inflammasome, decreased serum corticosterone levels, and increased 5-HT, NE, and BDNF concentrations in the hippocampus. These findings indicate that the NLGN3 gene may play a role in depression by influencing neurotransmitter levels and the NLRP3 inflammasome [28]. However, relatively few studies have investigated the NLGN3-NRXN axis in patients with EP and comorbid MDD, and the specific mechanism underlying the involvement of this axis remains unclear.

Single-cell RNA sequencing (scRNA-seq) facilitates the investigation of the com-position and heterogeneity of brain tissue cells, representing a crucial technique for elucidating the pathogenesis of epilepsy and depression and developing new therapeutic strategies [29]. Molecular dynamics simulation (MDs) and cell thermal stability migration assay (CETSA) provide valuable insights into the binding patterns and stability of protein-drug complexes, revealing the molecular mechanism of their interaction efficiently and comprehensively [30]. Therefore, this study aims to integrate methodologies such as scRNA-seq, MDs and CETSA to systematically investigate the mechanisms underlying interactions between key proteins and potential therapeutic agents for EP and comorbid MDD. This study provides novel ideas and strategies for the prevention and treatment of this condition.

## Materials and methods

### Data source

Data source: The integrated Gene Expression Omnibus (GEO) data repository (https://www.ncbi.nlm.nih.gov/gds) was searched for scRNA-seq data for MDD (GSE213982) and EP (GSE190452). The MDD dataset included 18 controls and 20 MDD patients, whereas the EP dataset included 4 controls and 4 EP patients. These samples were then combined for single-cell analysis.

### Cell clustering

The quality-controlled data were standardized by employing the normalize Data function of Seurat for R language-4.3.0. Cell-Cycle Scoring was used to calculate the cell cycle score, while high-variability genes were selected through the FindVariableFeatures function. The FindIntegrationAnchors and IntegrateData functions were utilized for data integration and batch effect elimination, whereas the ScaleData function was adopted to normalize the combined data. Furthermore, the data were subjected to principal component analysis (PCA; linear dimensionality reduction), and the FindClusters function was utilized to conduct unbiased clustering of cells and to visualize the dimensionality reduction in the form of uniform manifold approximation and projection (UMAP).

### Cell annotation

The FindAllMarker function was used to identify the genes that were significantly differentially expressed within each cell type in relation to the remaining genes. The cell types were annotated based on gene expression profiles in conjunction with references to published 10X scRNA-seq data and relevant research reviews on cell subpopulations.

### Analysis of cellular and signaling pathway interactions

Once the cells were annotated, the data were imported into CellChat for analysis to integrate the cellular communication networks visually. Chord diagrams, heatmaps, etc., were employed to present the number and strength of interactions between any two cell groups, as were signal input and output graphs. Violin plots and box plots were constructed to analyze the expression of proteins related to the NLGN3-NRXN pathway in OPCs from EP patients and MDD patients.

### Differentially expressed gene (DEG) analysis and protein–protein interaction (PPI) network construction

The FindMarkers function was used to identify DEGs within identical cell subpopulations (|foldchange| > 0.25, $P < 0.05$), and a differential gene expression map was produced. Overlapping DEGs were identified using a Venn diagram. The

interactions of the proteins encoded by the overlapping DEGs were analyzed using the STRING 12.0 online platform (https://string-db.org/). The generated PPI network was further visualized via Cytoscape 3.9.1 software.

## Spearman correlation analysis and gene set variation analysis (GSVA)

Spearman correlation analysis was used to identify the key genes strongly correlated with the key target proteins identified by CellChat analysis. Afterward, the gene sets were downloaded from the Molecular Signatures Database, and the GSVA algorithm was used to score each set to evaluate potential changes in biological function across samples.

## Target drug screening

The Connectivity Map platform (CMap, https://clue.io/) was used to identify drug candidates for EP and MDD. The upregulated and downregulated genes whose encoded proteins are localized to the plasma membrane in EP and MDD patients were uploaded to the CMap platform. Small molecules with low scores were selected as candidate drugs for alleviating EP and comorbid MDD.

## Molecular docking (MD) analysis and molecular dynamics simulation (MDs)

Drug structures were obtained from PubChem (https://pubchem.ncbi.nlm.nih.gov), a small molecule database, and protein structures were obtained from the PDB database (https://www.rcsb.org), a protein structure database. MD analysis was performed using AutoDock Vina (version 1.1.2), and the conformation with the highest affinity was selected for molecular dynamics simulation. In Gromacs (2024.03), a 100 ns full-atom MDS simulation was used for the MD complex, and a 2 fs step size and storing trajectory data every 10 ps were used for stability analysis, including root mean square deviation (RMSD), root mean square fluctuation (RMSF), radius of gyration (Rg), hydrogen bond analysis, Ramachandran analysis and principal component analysis (PCA) [31]. To understand how candidate compounds interact with proteins, stable trajectories from the 90–100 ns range of the later simulation stage were used to calculate the binding free energy of the ligand-protein complex; this was accomplished using the molecular mechanics generalized born surface area (MM-GBSA) method, which considers van der Waals interactions (VDWAALS), electrostatic interactions (EEL), gas phase free energy (GGAS), generalized born solvation energy (EGB), excess surface free energy (ESURF), solvation free energy (GSOLV), and total binding free energy (TOTAL).

## Cell thermal stability migration assay (CETSA)

All animal procedures were performed in strict accordance with the recommendations in the Guide for the Care and Use of Laboratory Animals of the National Institutes of Health, and the animal study protocol was approved by the Experimental Animal Management and Ethics Committee of North Sichuan Medical College (approval ID:2024052). C57 mice (20–24 g, male) obtained from the Experimental Animal Center of Huazhong University of Science and Technology (Hubei, China; License: SCXK 2023−0032) and housed under Specific Pathogen-Free (SPF) raising conditions. Surgical procedures were performed under sodium pentobarbital anesthesia, and all efforts were made to minimize animal suffering. Following anesthesia, mice (n = 3 per group) were euthanized by cervical dislocation, then transcardially perfused with phosphate-buffered saline. Hippocampal and Prefrontal brain tissue were rapidly dissected, flash-frozen in liquid nitrogen, and stored at −80°C for subsequent analysis. Mouse brain tissue was first homogenized, after which the total protein extract was obtained by centrifugation. The samples were divided into two groups: the experimental group, in which 30 µL of 100 µM/µL ziprasidone (A386746; Ambeed, Inc., Shanghai, China), with DMSO as the solvent, was coincubated with NLGN3 (29319−1-AP; Proteintech, Wuhan, China; 1:5000 dilution) protein at 37°C for 2 hours, and the control group, which received DMSO without the addition of the drug. Afterward, both groups were subjected to thermal treatment across nine temperature gradients (from 30°C to 94°C). The NLGN3 protein not bound to ziprasidone underwent denaturation and precipitated.

After centrifugation, the soluble proteins in the supernatant were collected, and the levels of NLGN3 protein were quantified using Western blot analysis. The greater retention of soluble NLGN3 in the experimental group at elevated temperatures indicates that ziprasidone significantly enhances the thermal stability of NLGN3 through direct binding.

The animal study protocol was approved by the Experimental Animal Management and Ethics Committee of North Sichuan Medical College (IACUC-2024052). All experimental protocols and experiments were in conformity with the Helsinki Declaration and ARRIVE guidelines.

### Western blot analysis

Total protein was extracted from the prefrontal brain tissue of C57 mice (n = 3) using RIPA lysis buffer (Beyotime, Shanghai, China). Protein concentrations were determined using a BCA kit (Vazyme, Nanjing, China). Equal amounts of protein were loaded onto 10% SDS–PAGE gels (EpiZyme, Shanghai, China) and subsequently transferred to PVDF membranes (Millipore, MA, USA). The membranes were blocked for 2 h at room temperature with 5% skim milk. Each membrane was then incubated overnight at 4°C with primary antibodies. Following incubation with primary antibodies against NLGN3, NLGN3+DMSO, NLGN3+ziprasidone+DMSO and β-actin (81115–1-RR; Proteintech, Wuhan, China; 1:20000 dilution), the membranes were washed three times with Tris-buffered saline with Tween 20 (TBST), incubated with goat anti-rabbit IgG (H+L) HRP (#SB-AB0101; ShareBio, Shanghai, China; 1:5000 dilution) secondary antibodies diluted for 2 h at room temperature, and then washed three times with TBST saline. Chemiluminescent detection reagents (Vazyme, Nanjing, China) were applied to visualize protein bands using a ChemiDoc MP system (Bio-Rad, CA, USA), and ImageJ software was used for image analysis.

### Data analysis

Statistical analysis was conducted via GraphPad Prism (8.0). Differences between two groups were analyzed via t test, Spearman correlation analysis was used to analyze the correlations of key genes, and comparisons between groups were conducted via one-way analysis of variance (ANOVA). The experimental data are expressed as the mean ± SD. $P < 0.05$ indicated a statistically significant difference.

## Results

### Single-cell profiling of EP and MDD patients

In this study, scRNA-seq data for brain tissues from EP (GSE190452) and MDD (GSE213982) patients included in the GEO database were evaluated. Considering the data quality of multiple samples comprehensively, cells with fewer than 200 captured genes were filtered out, and a total of 168,514 cells were retained (percent. MT < 15%). The known markers mentioned in the Seurat, CellMarker and other databases as well as in the literature were employed for cell clustering annotation. These cells were divided into 16 clusters via the UMAP algorithm and were annotated as excitatory neurons, inhibitory neurons, astrocytes, OPCs, oligodendrocytes, microglia, and neurovascular units (Fig 1A-1B). The marker genes of these 7 cell types were visualized, and the heatmap (Fig 1C) results demonstrated that the clustering effect of the marker genes was strikingly obvious, suggesting that the cell type identification was accurate.

### The number of cells and differential gene expression in EP and MDD patients

After defining the cell types, we utilized the t test method to determine whether differences existed and whether the differences in each cell subpopulation between patients and healthy individuals were significant. The cell percent ratio of the bar chart was used to depict differences in the composition ratios of each cell subpopulation between patients and healthy individuals in the EP and MDD datasets (Fig 2A). Compared with those in the respective control groups, the EP cell subpopulation was not significantly altered ($P > 0.1$), but the MDD cell subpopulation was significantly decreased in the number of astrocytes ($P = 0.00093$) and OPCs ($P = 0.0038$). Further analysis revealed obvious alterations in the numbers

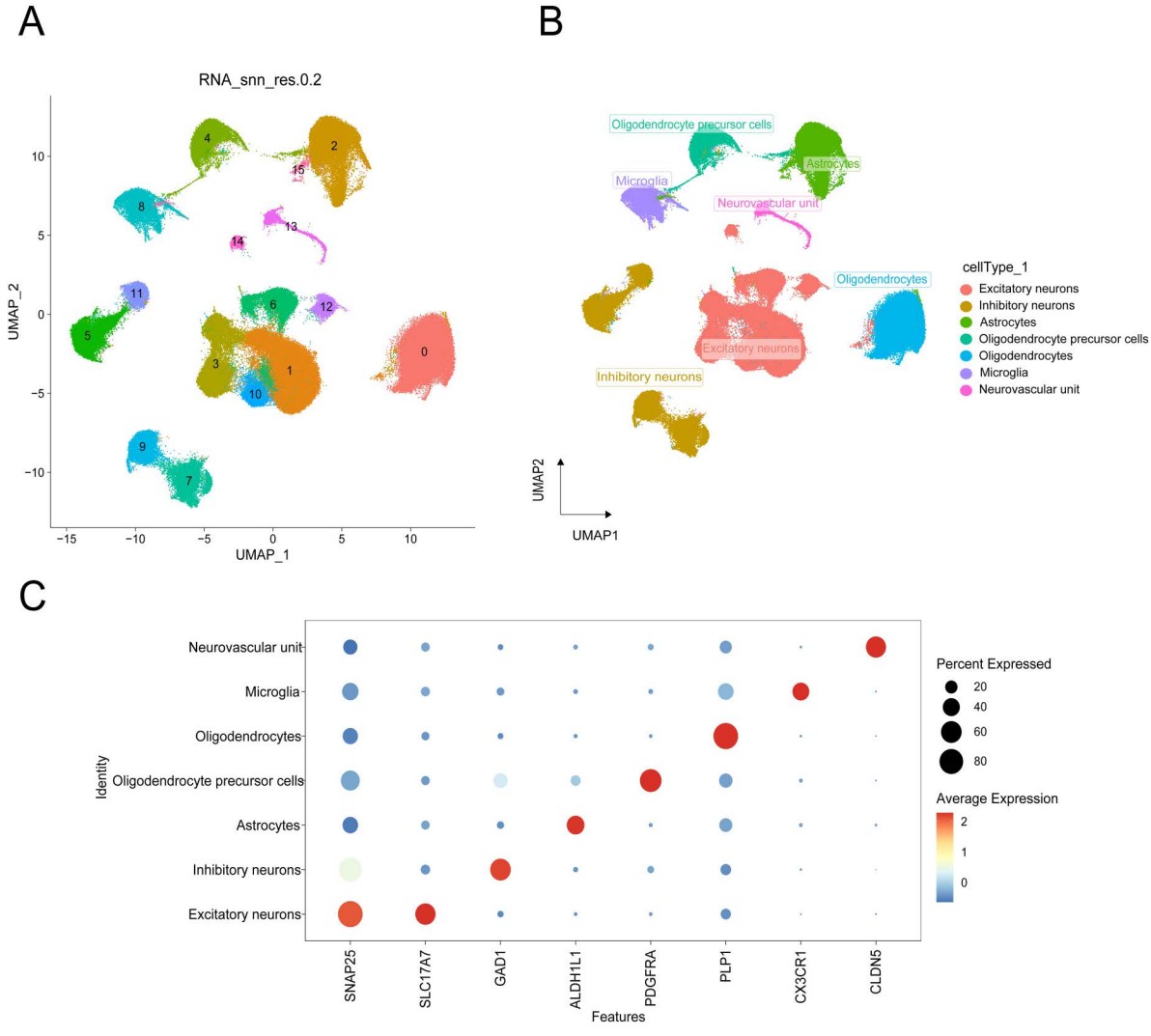

**Fig 1. Single-cell profiling of EP and MDD patient data. (A)** Cell clustering. **(B)** Cell types in EP and MDD patients. **(C)** Dot plot of cell marker genes across cell types.

of oligodendrocyte precursor cell subpopulations in the EP ($P = 0.1$) and MDD ($P = 0.0038$) patient groups (Fig 2B). The expression patterns of the DEGs in various cell subpopulations were subsequently determined. The number of differentially expressed genes in excitatory neurons was greatest in EP patients, whereas in MDD patients, the number of genes in inhibitory neurons was greatest. However, the most significant difference in the number of up-regulated and down-regulated genes was observed in the OPCs of EP and MDD patients (Fig 2C-2D). These findings, the significant disparity in the quantity of upregulated versus downregulated genes of OPCs and their differentially expressed genes in EP and MDD, suggest that OPCs may play an important role in EP with comorbid MDD.

### Differentially expressed gene analysis of OPC subsets in EP and MDD datasets

We subsequently performed an in-depth analysis of the DEGs across the OPC subpopulations. An integrative analysis of the OPC datasets revealed 276 upregulated and 166 downregulated DEGs in EP patients and 72 upregulated and

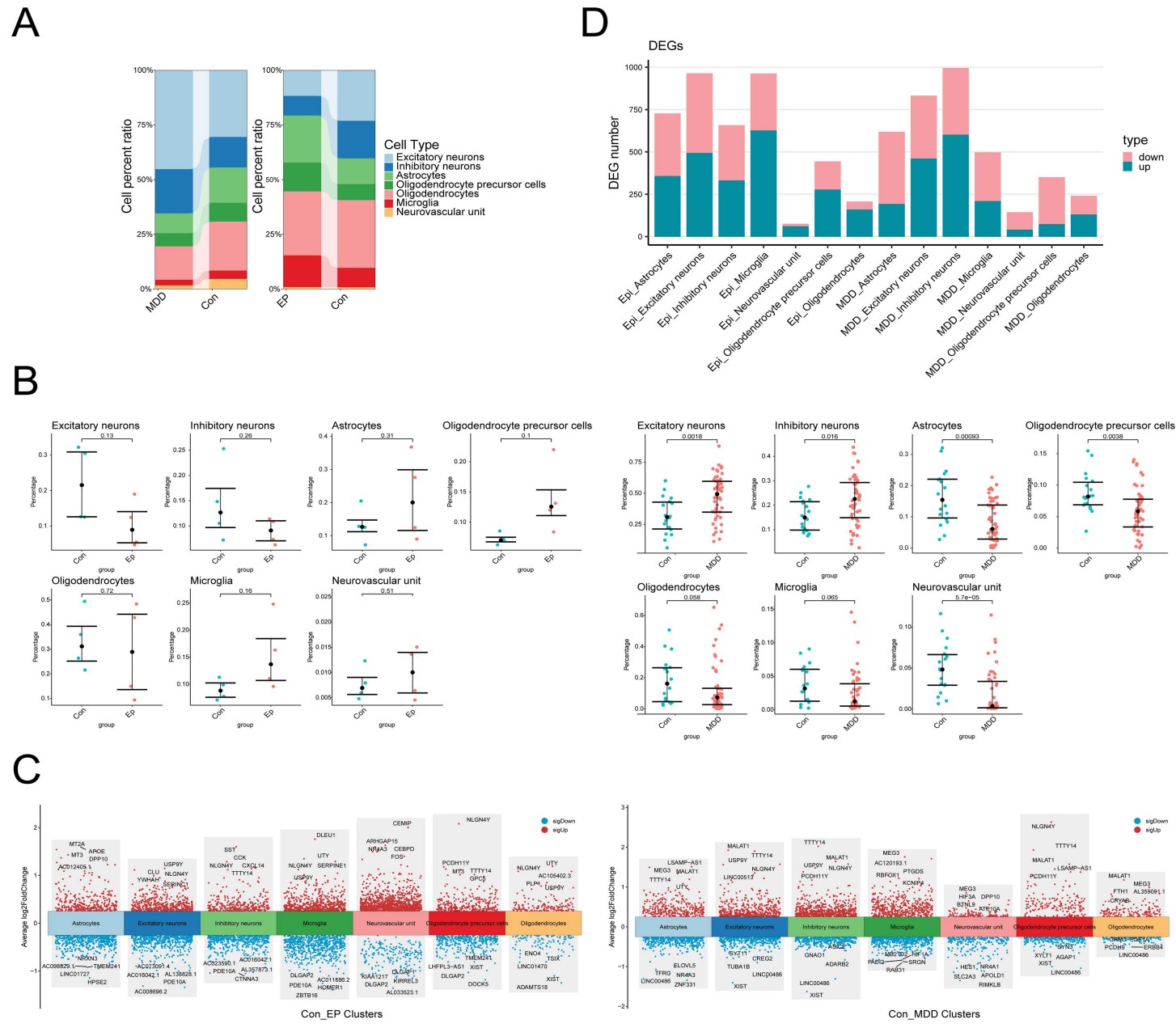

**Fig 2. Numbers of differentially expressed genes related to microglia in the EP and MDD datasets. (A)** Variations in the proportions of the cell types across samples. **(B)** Analysis of differences in cell number among the various groups. **(C)** Scatter plot of genes differentially expressed across cell types. **(D)** Bar chart of genes differentially expressed across cell types. The data are presented as the mean±SD, and $P < 0.05$ indicated statistical significance.

277 downregulated DEGs in MDD patients (Fig 3A). Venn diagrams revealed 39 genes that were commonly differentially expressed among OPC subsets (|Log2FC|>0.25, $P\_adj < 0.05$), 21 of which were jointly upregulated and 18 of which were jointly downregulated (Fig 3B). GO enrichment analysis revealed that the upregulated genes were associated primarily with pathways such as cytosolic ribosomes, ribosomes, and cytoplasmic translation, whereas the downregulated

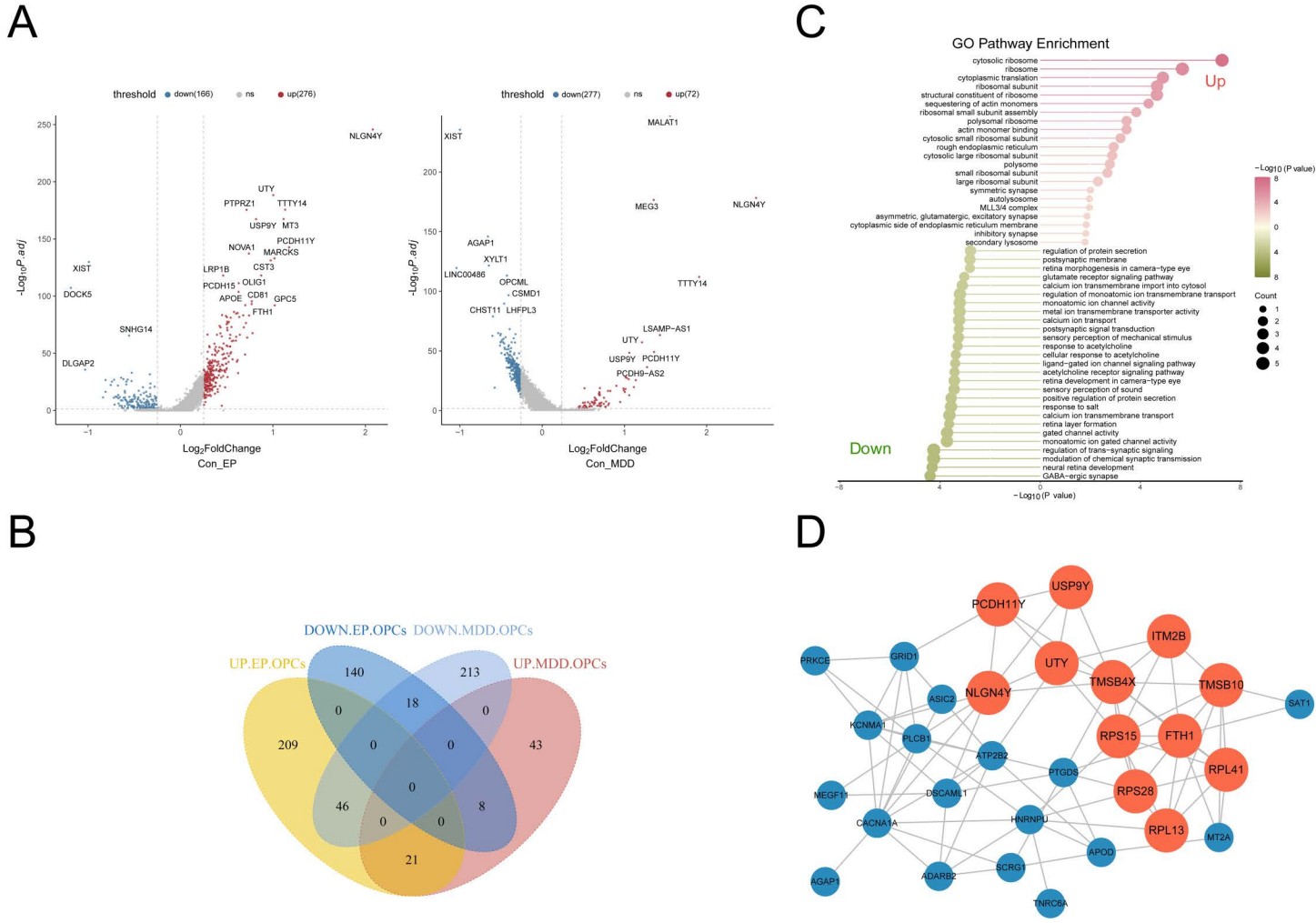

**Fig 3. DEG analysis of the OPCs in the EP and MDD datasets. (A)** Volcano map of DEGs. **(B)** Venn diagram of DEGs in OPCs. **(C)** GO enrichment analysis of the genes that were differentially co-expressed in OPCs. **(D)** PPI network of DEGs in OPCs, red represents the gene with the highest score.

genes were associated with pathways such as GABAergic synapses, neural retina development, and chemical synaptic transmission modulation (Fig 3C). Finally, using the MCODE function in Cytoscape, strong interactions were detected among 12 genes, namely, PCDH11Y, USP9Y, UTY, NLGN4Y, TMSB4X, ITM2B, TMSB10, FTH1, RPS15, RPS28, RPL41, and RPL13, forming a visual PPI network (Fig 3D).

### The NLGN3-NRXN pathway in OPCs regulates synaptic plasticity in EP and MDD

We utilized CellChat analysis to evaluate whether the NLGN3-NRXN pathway regulates synaptic plasticity in EP and MDD patients. The preponderance of interactions occurred between OPCs and excitatory neurons in both the EP group and the MDD group (Fig 4A); once the interaction intensity was accounted for, the interaction in-tensity of OPCs in the EP and MDD datasets increased (Fig 4B). Therefore, the OPCs interacted more frequently and strongly with other cell subsets.

To determine the communication signaling pathways between OPCs and other cell subpopulations, we computed the output and input intensities of each signaling pathway within the cell subpopulations. The NRXN signal emerged as the

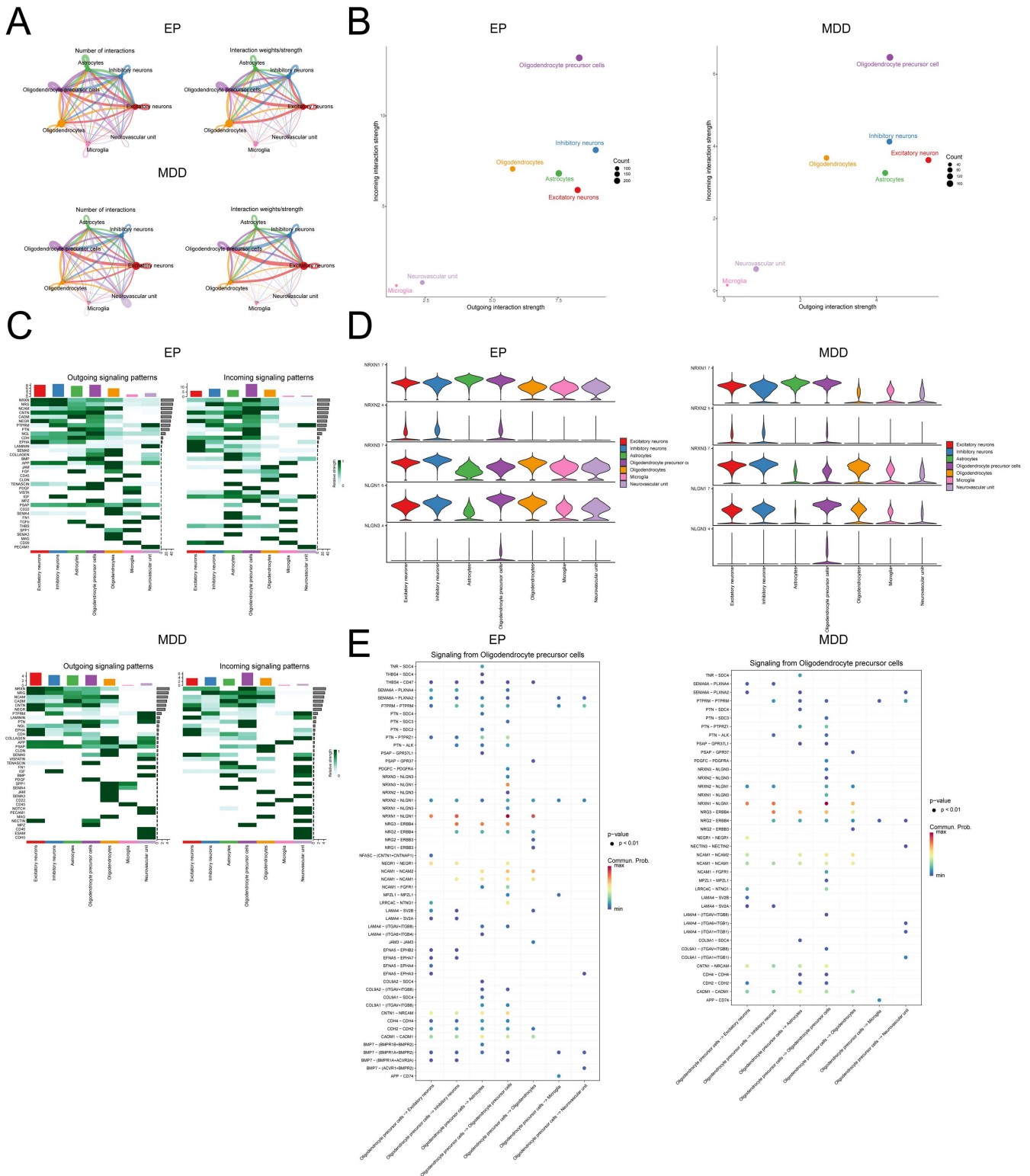

**Fig 4. Cellular interactions of cell subpopulations in EP and MDD patients. (A)** Cell-cell communication between cell types. **(B)** Total number of interactions between cell types in the EP and MDD groups. **(C)** Output and input intensities of signaling pathways within OPC subpopulations. **(D)** Violin plots of the expression of proteins in the NRXN signaling pathway. **(E)** Scatter plot of receptor-ligand signals emitted by OPCs interacting with cell subsets.

predominant output signal in OPCs in both EP and MDD patients (Fig 4C). The expression profiles of ligands and receptors in the NRXN pathway revealed that NLGN3 is expressed exclusively in OPCs in the EP and MDD datasets (Fig 4D-4E). Ligand-receptor bubble plots of OPCs revealed that these cells primarily signal to astrocytes (Fig 4E). Recent research has revealed that the NRXN signaling pathway is essential for synapse structure and functional regulation in the nervous system; it binds with neural ligand proteins (NLGN1/2/3/4) to create cross-synaptic adhesion complexes, influencing synaptic formation, specificity, and remodeling [21]. Based on these findings, we hypothesize that the NLGN3 protein in OPCs facilitates the regulation of synaptic function by astrocytes via the NRXN signaling pathway, thereby playing a significant role in the pathophysiology of EP and MDD.

## Key genes linked to the NLGN3 protein are involved primarily in immune-inflammatory path-ways such as oxidative phosphorylation and mTORC1 signaling

Spearman correlation analysis was used to examine the relationships between the expression levels of the target protein NLGN3 and those of the 12 differentially expressed genes identified through the PPI network in OPCs. The results revealed that the expression of three genes, RPS28, TMSB10 and TMSB4X, was positively correlated with NLGN3 expression ($P < 0.01$, $R > 0.8$) (Fig 5A). These three genes were identified as key genes. To elucidate the potential molecular mechanisms through which these genes influence the progression of EP and MDD, we subsequently analyzed the signaling pathways enriched by these three hub genes. The GSVA results indicated that when highly expressed, RPS28 primarily activates immune-inflammatory and gene transcription signaling pathways such as oxidative phosphorylation, mTORC1, and MYC targets v1 (Fig 5B). Moreover, high expression of TMSB10 and TMSB4X primarily activates immune-inflammatory and DNA repair pathways, oxidative phosphorylation, mTORC1, and DNA repair. When these genes are downregulated, they predominantly activate the immune-inflammatory-associated IL-6-JAK-STAT3 signaling pathway, bile acid metabolism, and the UV response (DN) signaling pathway (Fig 5C-5D). Finally, we used a t test to compare the expression of three key proteins (RPS28, TMSB10, and TMSB4X) between the control and disease groups in the OPC subgroup and found that all three proteins were highly expressed in the disease group (Fig 5E). These results indicate that key genes associated with the NLGN3 protein primarily participate in immune-inflammatory pathways, including oxidative phosphorylation and mTORC1 signaling, influencing the progression of EP and MDD.

### Identification of target drugs

CMap was used to identify drug candidates for the treatment of EP and MDD. If a small molecule had a negative score, it was considered potentially effective, and greater negative scores indicated greater efficacy. Six candidate drugs with relatively high negative values were selected (Table 1).

### Ziprasidone can effectively bind to the key protein NLGN3

We subsequently conducted molecular docking (MD) analyses of NLGN3 and the most significantly correlated gene, RPS28 ($R = 0.83$, $P < 0.01$), against six selected candidate drugs that exhibited relatively high negative scores. The binding modes with the highest affinities for Ziprasidone with NLGN3 (−7.5 kcal/mol) were analyzed (Table 2). All the small-molecule drugs could bind to the protein pocket, with good shape matching, and could maintain binding with the protein through multiple forces, such as hydrogen bonds and van der Waals forces (Fig 6A-6I). Ziprasidone was located within the hydrophobic cavity formed by the residues VAL628, PRO629, ALA317, LEU318, TYR441 and LEU440 of NLGN3; it was capable of forming 2 hydrogen bonds simultaneously with TYR632 and GLU485, resulting in the most favorable binding interaction between the two (Fig 6D-6F).

### Ziprasidone stably interacts with the core protein NLGN3

MDs were used to assess the binding and stability of the NLGN3-ziprasidone complex. For the 100 ns simulations, the RMSD values for the protein backbone, ligand, and binding site remained at approximately 1 Å, indicating a stable

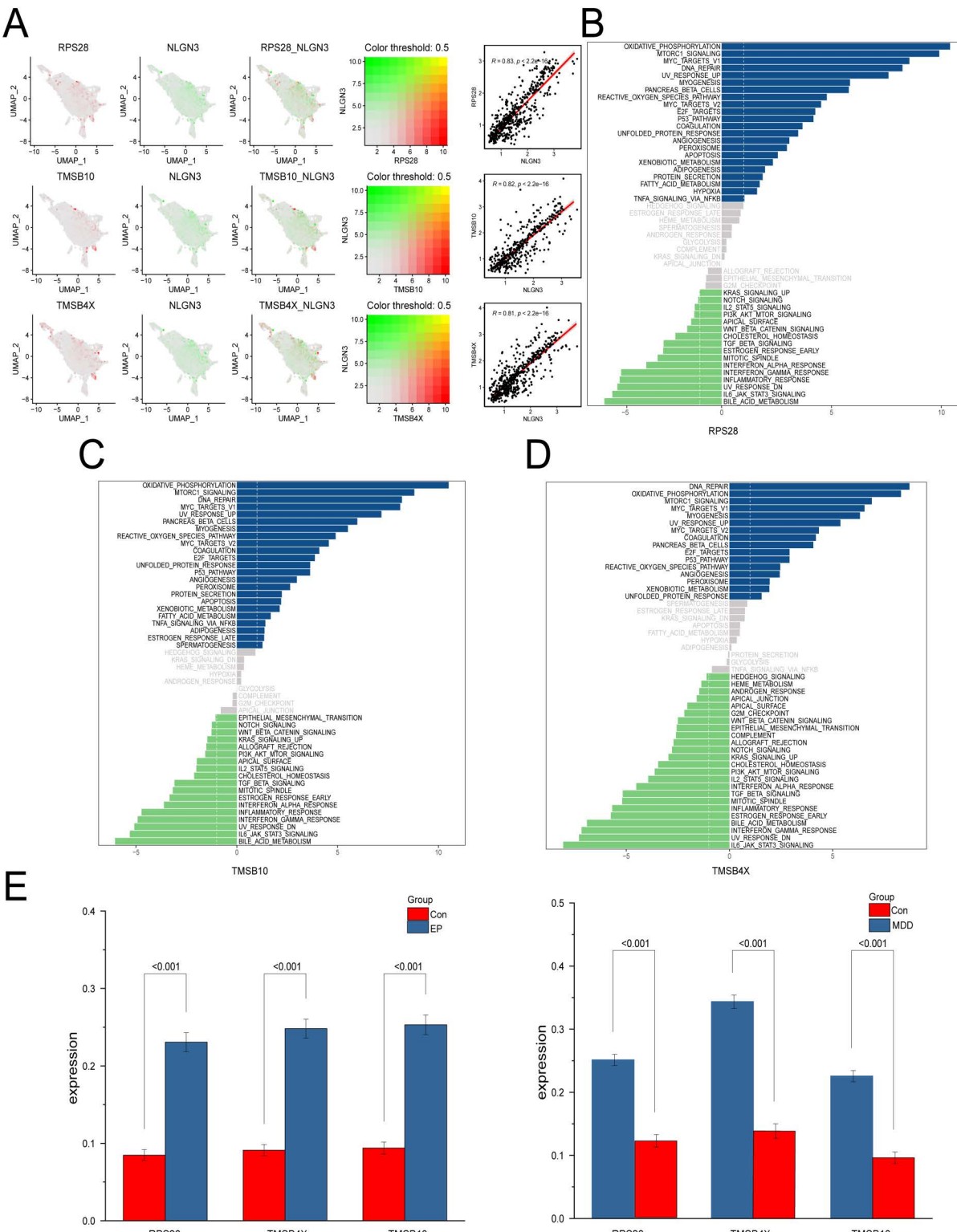

**Fig 5. The key genes linked to the NLGN3 protein are involved primarily in immune-inflammatory pathways. (A)** Spearman correlation analysis between NLGN3 and key genes. **(B-D)** GSVA of key genes: blue represents pathways linked to high gene expression, and green represents pathways linked to low gene expression, while gray indicates non differentially expressed pathways. **(E)** Bar chart of the key DEGs. The data are presented as the mean±SD, and *P*<0.05 indicated statistical significance.

**Table 1. Identification of drug candidates for the treatment of EP and MDD with CMap.**

| Rank | Potential drug_id | Potential drug name | Score |
|------|-------------------|---------------------|-------|
| 1 | BRD-K15014948 | Tranexamic acid | −1.7388 |
| 2 | BRD-K52172416 | Anastrozole | −1.7302 |
| 3 | BRD-K29582115 | Ziprasidone | −1.7107 |
| 4 | BRD-K09485525 | GANT-61 | −1.7061 |
| 5 | BRD-K09397065 | SR-57227A | −1.7029 |
| 6 | BRD-K80778372 | Ro-19–4605 | −1.6899 |

**Table 2. Results of MD between key target proteins and active compounds.**

| Protein | Ligand | Binding energy (kcal/mol) | Residue(s) involved in hydrogen bonds |
|---------|--------|---------------------------|----------------------------------------|
| NLGN3 | Tranexamic acid | −4.5 | GLU485 |
| | Ziprasidone | −7.5 | GLU485, TYR632 |
| RPS28 | Tranexamic acid | −3.963 | GLU60 |

conformation (Fig 7A). The mean RMSF values for NLGN3 and ziprasidone were 1.70 Å and 1.65 Å, respectively, indicating high ligand stability in the binding pocket (Fig 7B). The Rg values for both the protein (23.16 Å) and the ligand (4.37 Å) exhibit minimal fluctuation, suggesting a relatively compact and stable structural conformation (Fig 7C). Hydrogen bond analysis revealed that despite the relatively low number of hydrogen bonds formed in the protein-ligand complex, the majority of these bonds possess favorable geometric characteristics and demonstrate high structural stability. Notably, the key residues GLU485 and TYR632 serve as critical components within the hydrogen bond network, supporting previous docking results (Fig 7D). MM-GBSA analysis revealed a strong binding free energy of −33.83 kcal/mol, driven mainly by EEL, with GGAS outweighing GSOLV. The critical residue, glutamine 485, potentially exerted a detrimental influence on the binding process, as indicated by its positive energy value (Fig 7E). Ramachandran analysis revealed that the secondary structure distribution of the protein-ligand complex was balanced and stable during the simulation process (Fig 7F). PCA revealed the main movement patterns of the protein-ligand complex by showing its distribution in the first two principal components (PC1 and PC2) space, indicating the lowest free energy points and characteristic conformations, and providing an intuitive view of the energy distribution of the system (Fig 7G-7H). Overall, the analysis confirmed the stable structure of the ziprasidone-NLGN3 complex, with MM-GBSA energy analysis supporting thermodynamic binding stability.

## Validation of targeted binding between ziprasidone and the NLGN3 protein via CETSA

The direct binding properties of the candidate small-molecule drug ziprasidone to the key target protein NLGN3 were confirmed using CETSA. Experimental data demonstrated that within the temperature gradient range of 30–94°C (Fig 8A), the thermal stability of the NLGN3 protein in the ziprasidone-treated group was significantly greater than that in the solvent control group (DMSO group; $P = 0.0125$) (Fig 8B-8C). These results indicated that ziprasidone forms stable complexes through specific binding to NLGN3, effectively enhancing its thermal resistance.

## Discussion

EP and MDD are highly comorbid, particularly in temporal lobe epilepsy, with approximately 30% of temporal lobe epilepsy patients experiencing both conditions [2, 32]. Moreover, the complex, reciprocal relationship between epilepsy and depression can lead to a vicious cycle in which depression exacerbates the frequency and severity of seizures [33]. Therefore, novel treatment strategies to improve the quality of life and survival rate of patients with EP and comorbid MDD

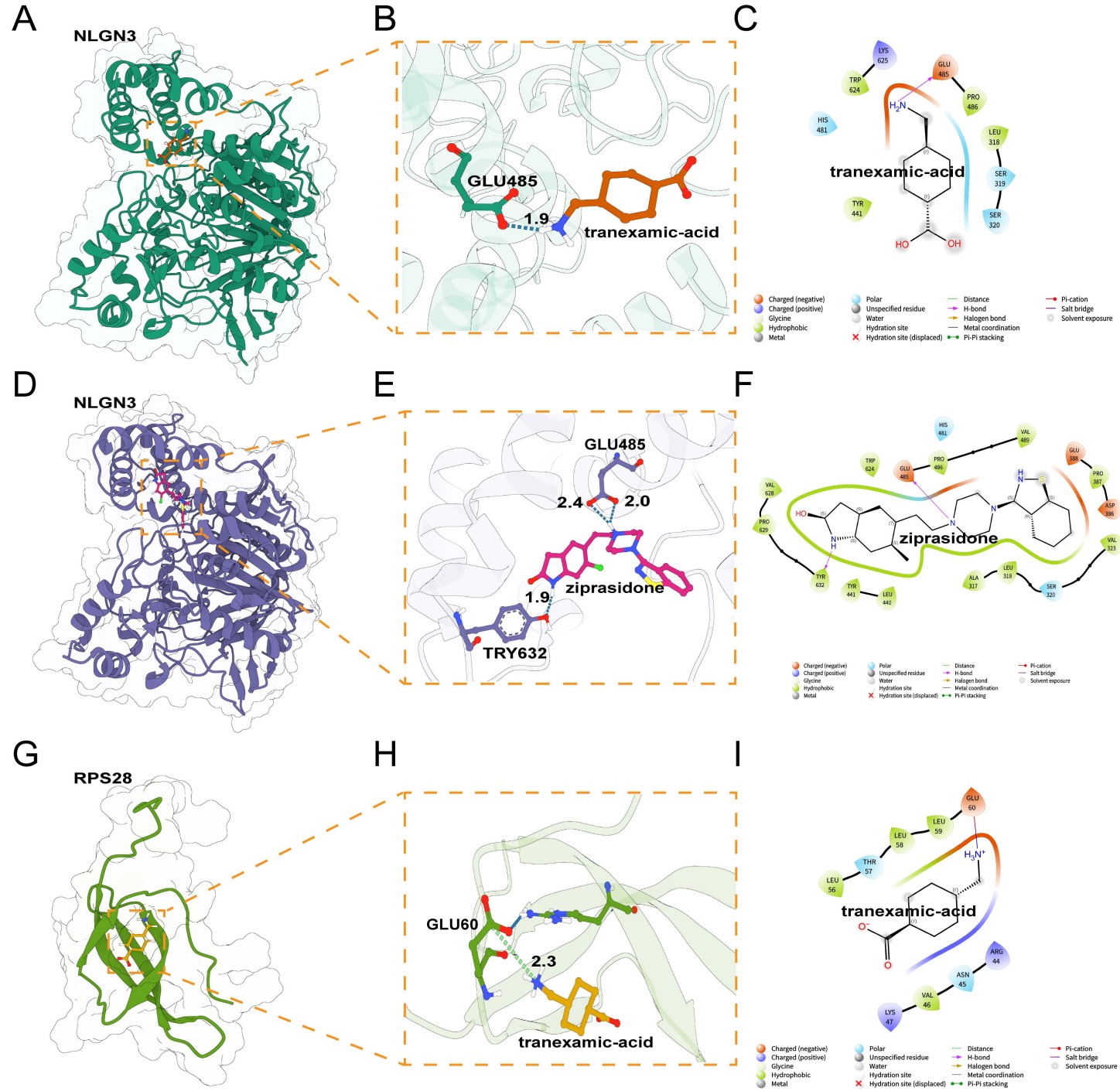

**Fig 6. Molecular docking analyses of key proteins and candidate drugs. (A, D, G)** 3D structures of protein–small molecule complexes, with boxes indicating binding pockets. **(B, E, H)** 3D interactions within these pockets. **(C, F, I)** Chemical structures and 2D interactions of small molecules with nearby amino acids.

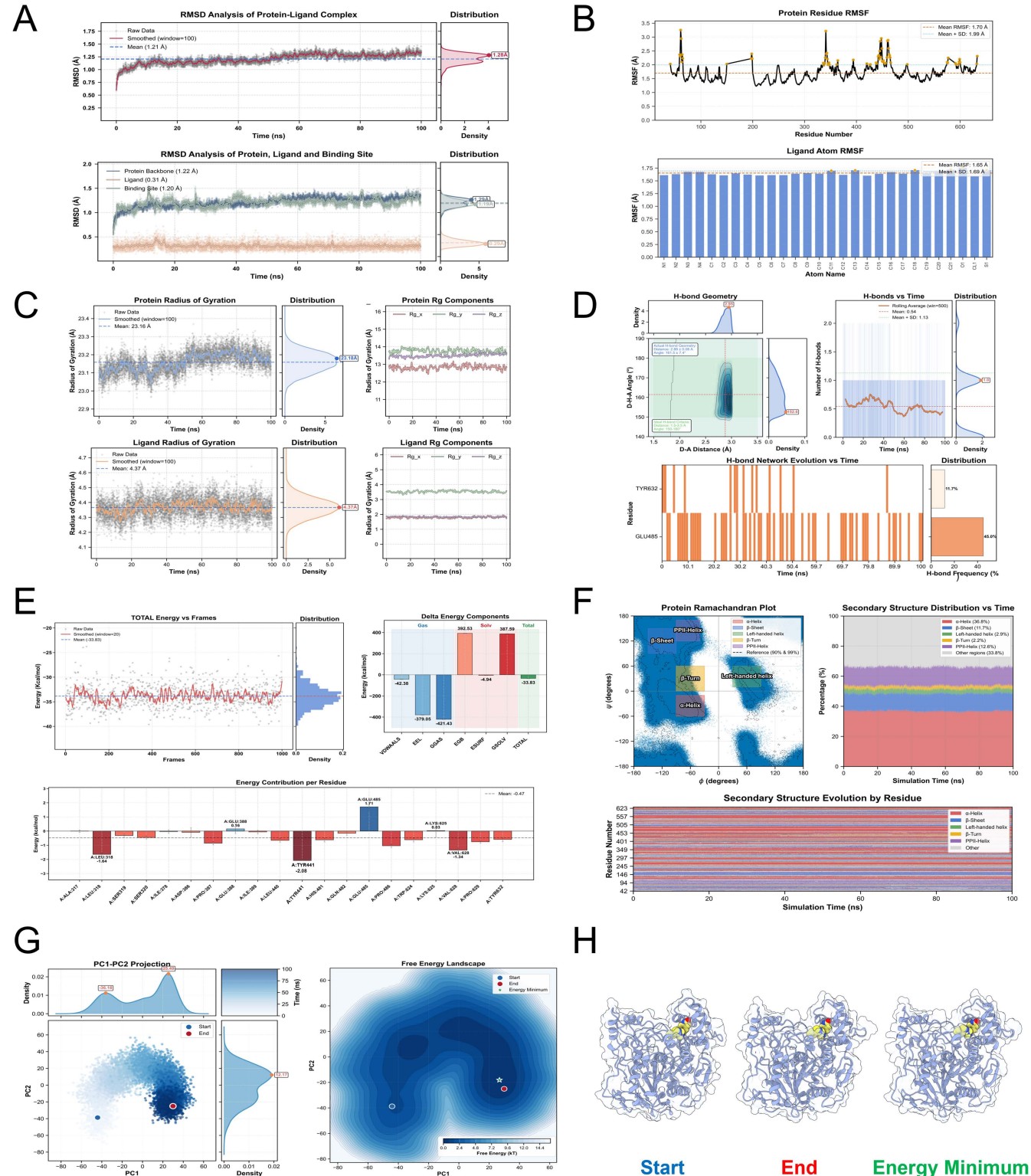

**Fig 7. MDs of the key gene NLGN3 and the candidate drug ziprasidone. (A)** RMSD. **(B)** RMSF. **(C)** Rg analysis. **(D)** Hydrogen bond analysis. **(E)** Binding free energy analysis. **(F)** Ramachandran analysis. **(G)** PCA. **(H)** 3D conformations corresponding to the start point, end point and energy minimum in the G diagram.

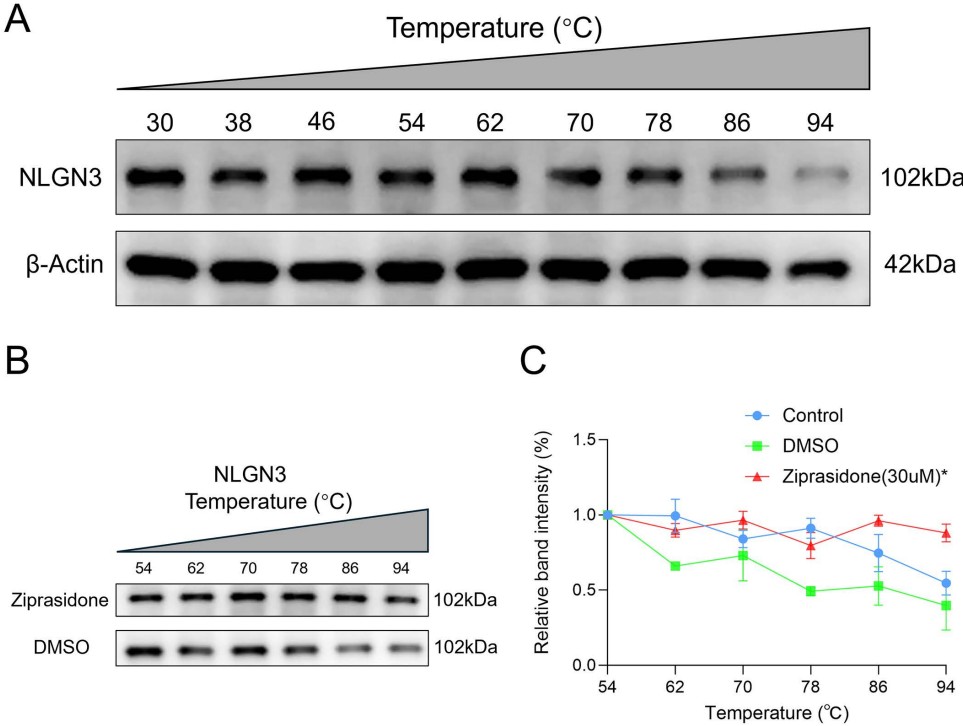

**Fig 8. CETSA was used to detect the interaction between ziprasidone and NLGN3. (A)** The expression of NLGN3 under different temperature treatments. **(B)** NLGN3 protein expression at different temperatures after ziprasidone treatment. **(C)** Comparison between the DMSO group and the ziprasidone group. n = 3 independent experiments, * $P < 0.05$.

---

are urgently needed. In this study, we used scRNA-seq to investigate cellular interactions, and through MDs and CETSA experiments, we evaluated the binding stability and interaction mechanism of key proteins with the best candidate drugs. Our findings elucidate the effect of ziprasidone on the NLGN3-NRXN signaling pathway in OPCs. These results not only confirmed the direct interaction between ziprasidone and NLGN3 but also elucidated its potential mechanism of action: the drug may stabilize the conformational structure of the NLGN3 protein, activate the NLGN3-NRXN signaling pathway of OPCs, and subsequently regulate synaptic remodeling capability through the suppression of the neuroinflammatory response.

Here, we identified cell types involved in EP and MDD, observing changes primarily in the number of OPCs, along with neurons, astrocytes and microglia. Previous single-cell sequencing analyses of MDD patients revealed significant dysregulation of deep excitatory neurons, OPCs, and astrocytes [34–36]. These findings are essentially consistent with the results of the scRNA-seq of samples from MDD patients in our study. Most of the previous reports focused on microglial activation and increases in the number of microglia in MDD [37, 38]. However, our findings indicate a reduction in the number of microglia. This discrepancy might be attributed to the fact that the sequencing samples included in this research were from female patients; however, microglial activation in women with MDD was lower, and the number of synaptic connections was greater [36, 39]. Consequently, future studies should focus on sex-specific transcriptome differences in MDD. In contrast, EP patients, particularly those with temporal lobe epilepsy, typically present an increase in the number of glial cells (mainly OPCs and astrocytes) and a notable decrease in the number of neurons, especially inhibitory ones [40, 41]. Our current research also confirmed these results.

Excess or insufficient synaptic pruning or dysregulation of this process is a key mechanism underlying depression and epileptic seizures [42, 43]. OPCs can remove excess synapses via phagocytosis and work with microglia and astrocytes to achieve synaptic pruning and neural circuit refinement [15–17]. Additionally, under certain conditions, OPCs and glial cells can interact and differentiate into astrocytes [9, 12]. Our study revealed that OPC subsets interact primarily via ligand-receptor pairs involved in the NLGN3-NRXN signaling pathway and that OPCs send the greatest number of signals to astrocytes. NLGN3-αNRXN1 signaling can specifically regulate GABAergic synaptic function in the hippocampus of mice [44]. Astrocytic NLGN3 protein can regulate social memory and synaptic plasticity via adenosine signaling in male mice [45]. Therefore, we speculate that the NLGN3-NRXN signaling pathway in OPCs mediates the effects of astrocytes on synaptic structure and function, inhibits neuroinflammation, and thus plays a crucial role in EP and MDD.

Notably, deleting astrocytic NLGN3 alters gene expression and mouse behavior without affecting cerebellar synapse number or function, suggesting that astrocytic NLGN3 is not involved in trisynapses but may regulate gene expression [46]. Moreover, the complete absence of astrocyte neuro connections (NIGN1/2/3) does not affect the number of synapses, synaptic function or astrocyte morphology in the cortex [47]. These studies indicate that the NIGN3 protein, which controls synaptic structure and function, likely originates from neurons and OPCs [24]. Our scRNA-seq sequencing study also confirmed that the NLGN3 protein mainly expressed in OPCs.

Further analysis revealed that three key genes, RPS28, TMSB10, and TMSB4X, were significantly positively correlated with the protein NLGN3 in the OPC subgroups of EP and MDD patients. These genes are involved primarily in immune inflammatory pathways, such as oxidative phosphorylation and mTORC1 signaling, which are crucial in EP and MDD. mTOR signaling plays a crucial role in regulating neural processes such as stem cell activity, neural development, and synaptic plasticity. Overactivation of the mTORC1 pathway is linked to conditions such as epilepsy, anxiety, and sleep disorders [48]. Targeting the mTORC1 substrate S6K1 can improve synaptic issues and reduce seizures and depression in rats with chronic epilepsy [7]. Oxidative phosphorylation is crucial for mitochondrial function, and its dysfunction leads to lipid accumulation in astrocytes and severe neuroinflammation, which can worsen temporal lobe EP [49]. MDD is linked to mitochondrial dysfunction and altered proinflammatory cytokine expression, which may drive its pathogenesis [50]. The above research results are consistent with our findings.

Our MD study also revealed that ziprasidone strongly bound to NLGN3, forming two hydrogen bonds (TYR632 and GLU485) with a binding energy of −7.5 kcal/mol. MDs confirmed the stable interaction between the ziprasidone ligand and NLGN3 protein. Ziprasidone, developed by Pfizer and launched in 2002, is an effective 2-indoleone antipsychotic used primarily to treat schizophrenia, bipolar disorder, and MDD [51]. This second-generation antipsychotic effectively treats schizophrenia by uniquely acting as a high affinity antagonist of dopamine D2 receptors and an agonist of 5-HT 1A receptors [52]. Monoamines, such as 5-HT, DA, and NE, influence the seizure threshold and pattern of epilepsy by modulating neuronal excitability and inhibition. In cases where epilepsy and depression coexist, serum 5-HT levels are significantly reduced, and this decrease is closely linked to depression [53]. Moreover, an increase in 5-HT levels can trigger long-term inhibition (LTD), weakening synaptic strength and impacting the development of synaptic plasticity [54]. An imbalance between dopamine and 5-HT levels may be crucial in epileptic seizures, emphasizing the role of neurotransmitters in synaptic plasticity and nervous system balance [55, 56]. These studies suggest that ziprasidone may alter serotonin, dopamine, and norepinephrine levels, remodel synaptic plasticity by NLGN3-NRXN signaling pathway, providing therapeutic benefits for EP and MDD patients.

## Conclusion

In summary, through the integration of scRNA-seq data from patients with EP and MDD, we conducted cell communication analysis and reported that the frequency and intensity of interactions between OPCs and other cells significantly increased. NLGN3-NRXN signaling was identified as the primary communication pathway between OPCs and astrocytes. Furthermore, the CMap database was used to select the top 6 small-molecule drugs. Next, through MDs and CETSA

experiments, we evaluated the binding stability and interaction mechanism of key proteins with the best candidate drugs. The results confirmed that NLGN3 formed stable complexes through specific binding to ziprasidone. The stable complex might activate the NLGN3-NRXN signaling pathway, influencing astrocyte synaptic remodeling and reducing neuroinflammation. Despite certain unresolved challenges, this study provides insights into the molecular interactions between ziprasidone and NLGN3 and the underlying mechanisms, suggesting that ziprasidone may influence the pathophysiology of EP and MDD to validate the way for the development of more effective therapeutic strategies.

## Supporting information

**S1 File. Uncropped images for all blot results.**
(PDF)

**S2 File. MDs between ziprasidone and NLGN3.**
(MP4)

## Author contributions

**Conceptualization:** Guohui Jiang, Shushan Zhang.

**Formal analysis:** Guiqin Bai, Xuerong Zhou.

**Funding acquisition:** Guiqin Bai, Dazhang Bai, Shushan Zhang.

**Investigation:** Dazhang Bai, Peilin Zhao, Tao Peng, Cheer Muer.

**Methodology:** Cheng Xiong.

**Project administration:** Xuerong Zhou.

**Supervision:** Shushan Zhang.

**Validation:** Xi Kang, Ruiqi Huang.

**Writing – original draft:** Xuerong Zhou.

**Writing – review & editing:** Guiqin Bai, Shushan Zhang.

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
