## [Decision Letter · Decision Letter 0]

23 Feb 2026

Dear Dr. Bai,

We look forward to receiving your revised manuscript.

Kind regards,

Peng Zhong, Ph.D.

Academic Editor

PLOS One

Journal Requirements:

“This study was supported by the Doctoral Start Fund of North Sichuan Medical College (CBY22-QDA18), the Special Project for Basic Research in Traditional Chinese Medicine of Sichuan Province (25MSZX560), the Scientific Research Development Project of the Clinical Medical College Affiliated Hospital of North Sichuan Medical College (2023PTZK012), the Sichuan Science and Technology Program (2023NSFSC0709, 2024NFSC0490), the Guangdong Basic and Applied Basic Research Foundation (2023A1515110847), and the Jiangmen Basic and Applied Basic Key Projects (2320002001026).”

3. We note that your Data Availability Statement is currently as follows: “All relevant data are within the manuscript and its Supporting Information files.”

Reviewers' comments:

Reviewer's Responses to Questions

**Comments to the Author**

1. Is the manuscript technically sound, and do the data support the conclusions?

Reviewer #1: Partly

Reviewer #2: Yes

2. Has the statistical analysis been performed appropriately and rigorously?

Reviewer #1: N/A

Reviewer #2: Yes

3. Have the authors made all data underlying the findings in their manuscript fully available?

Reviewer #1: Yes

Reviewer #2: Yes

4. Is the manuscript presented in an intelligible fashion and written in standard English?

Reviewer #1: Yes

Reviewer #2: Yes

Reviewer #1: The manuscript presents an interesting topic; however, it cannot be adequately evaluated in its current form due to the uniformly poor quality of all figures. All figures throughout the manuscript are severely blurred and lack sufficient resolution. Critical elements such as axis labels, cell-type annotations, pathway names, legends, and heatmap details are not legible even when magnified. As a result, it is impossible to independently assess the data or verify the authors’ interpretations.

Due to the inability to assess the data and conclusions based on the provided figures, I am unable to provide a meaningful scientific evaluation of the work. I therefore recommend rejection of the manuscript in its current form, with the suggestion that the authors resubmit after ensuring that all figures meet acceptable resolution and clarity standards.

Reviewer #2: Accept in present form

As the results suggests NLGN3 protein interacts with ziprasidone to form stable complexes, which may activate the

NLGN3-NRXN signaling pathway in OPCs and enhance synaptic remodeling by reducing

neuroinflammatory responses

.

Reviewer #1: No

Reviewer #2: **Yes:** Saumya PatelSaumya PatelSaumya PatelSaumya Patel

---

## [Author Response · Author response to Decision Letter 1]

21 Mar 2026

Dear Dr. Peng Zhong, Academic Editor, and esteemed reviewers

Thank you very much for giving us an opportunity to revise our manuscript, we appreciate dear editor and reviewers very much for your constructive comments and suggestions on our manuscript (PONE-D-25-56676). Based on your suggestion and request, we have tried our best to address each of the issues raised by improving the quality of all figures or by providing further discussion and clarification in the revised text. The academic editor and reviewer's suggestions or comments are italicized and numbered, with key points highlighted in blue. Our responses are in regular font, and any manuscript changes or additions are shown in red text. And we have carefully considered each point raised and have made the following revisions to our manuscript.

academic editor

Suggestion 1:

Submit a revised version of the manuscript that addresses the points raised during the review process. Please include the following items:

A letter labeled ‘Response to Reviewers’.

A marked-up copy of your manuscript labeled ‘Revised Manuscript with Track Changes’.

An unmarked version of your revised paper labeled ‘Manuscript’.

Response: Thank you very much for the recognition and professional suggestions on this study. In accordance with your guidance, we have uploaded the following three files: A “Response to Reviewers” letter, a marked-up version of the manuscript labeled “Revised Manuscript with Track Changes”, and a clean copy of the revised manuscript labeled “Manuscript”. The revised manuscript is now ready for your further consideration.

Suggestion 2:

Response: Thank you very much for your important comment. We have revised the manuscript and file names according to the template provided by PLOS ONE, to ensure that the manuscript complies with the formatting requirements of PLOS ONE.

Suggestion 3:

Please state what role the funders took in the study. If the funders had no role, please state: “The funders had no role in study design, data collection and analysis, decision to publish, or preparation of the manuscript.” Please include this amended Role of Funder statement in your cover letter.

Response: Thank you for your guidance. We have updated the Role of Funder statement as requested and included it in our cover letter. The statement is as follows:

“The funders had no role in study design, data collection and analysis, decision to publish, or preparation of the manuscript.” Please let us know if you require any further information or adjustments. We appreciate your assistance in updating the submission form.

Suggestion 4:

We note that your Data Availability Statement is currently as follows: “All relevant data are within the manuscript and its Supporting Information files.” If data are owned by a third party, please indicate how others may request data access.

Response: Thank you for your note regarding the Data Availability Statement. In this study, we mainly focused on analyzing publicly available datasets, which can be found in the GSE213982 (https://www.ncbi.nlm.nih. gov/geo/query/acc.cgi/acc=GSE 213982), and GSE190452 (https://www.ncbi.nlm.nih. gov/geo/query/ acc.cgi/acc=GSE 190452) datasets.

Suggestion 5:

PLOS ONE now requires that authors provide the original uncropped and unadjusted images underlying all blot or gel results reported in a submission’s figures or Supporting Information files. In your cover letter, please note whether your blot/gel image data are in Supporting Information or posted at a public data repository.

Response: Thank you for your guidance. We confirm that the original uncropped and unadjusted blot/gel image data underlying the results reported in this submission are provided in the Supporting Information files. In our cover letter, we have indicated that all original, uncropped, and unadjusted blot/gel image data are provided in the Supporting Information files.

Suggestion 6:

Your ethics statement should only appear in the Methods section of your manuscript. If your ethics statement is written in any section besides the Methods, please delete it from any other section.

Response: Thank you for your valuable comments on our manuscript. In accordance with your suggestion, we have relocated the ethics statement to the Methods section, and deleted it from any other section.

Suggestion 7:

Response: This point is not applicable, as the reviewers' comments did not include any specific recommendations to cite previously published works.

Reviewer #1

Comment 1:

The manuscript presents an interesting topic; however, it cannot be adequately evaluated in its current form due to the uniformly poor quality of all figures. All figures throughout the manuscript are severely blurred and lack sufficient resolution. As a result, it is impossible to independently assess the data or verify the authors’ interpretations.

Response: Thank you for your support and feedback on our manuscript. We apologize for the inconvenience caused by the suboptimal figure quality during your review. In accordance with your comments and the official figure preparation guidelines provided by PLOS ONE ( https://journals.plos.org/plosone/s/figures# loc-tools-for-figure-preparation), we have thoroughly revised and verified all figures to ensure full compliance. All figures have been modified as follows.

We have submitted all figures in high-resolution TIFF format as per your comments and journal's requirements. To ensure optimal evaluation, we kindly ask that, upon reviewing the revised manuscript, please download the original high-resolution figure files by clicking the “Download” link located in the upper-right corner of each figure placeholder.

Comment 2:

Is the manuscript technically sound, and do the data support the conclusions?

“Partly”

Response: Thank you very much for your important comment. Building on previous research [29-30], this study integrates single-cell RNA sequencing, molecular dynamics simulation, and cell thermal stability migration assay (CETSA) to systematically explore the mechanisms of interactions between key proteins and potential therapeutic agents for EP and comorbid MDD. Data source: The Gene Expression Omnibus (GEO) data repository (https://www.ncbi.nlm.nih.gov/gds) was searched for scRNA-seq data on MDD (GSE213982) and EP (GSE190452). The MDD dataset comprised 18 controls and 20 patients, while the EP dataset included 4 controls and 4 patients. Quality-controlled data were standardized using Seurat's normalize Data function in R 4.3.0. Proteins were extracted from hippocampal tissue of C57BL/6 mice (n = 3 per group) for cellular thermal shift assay (CETSA)-based target engagement validation. The original underlying images for all blot or gel data reported had been provided in our submission. Accordingly, all experiments were conducted rigorously under controlled conditions, with appropriate experimental controls, technical and biological replicates, and statistically justified sample sizes; conclusions were drawn strictly in accordance with the empirical evidence.

Comment 3:

Has the statistical analysis been performed appropriately and rigorously?

“N/A”

Response: Thank you very much for your precious suggestion. Statistical analysis was conducted via GraphPad Prism (8.0). Differences between two groups were analyzed via t test, Spearman correlation analysis was used to analyze the correlations of key genes, and comparisons between groups were conducted via one-way analysis of variance (ANOVA). For further information, please consult the data analysis subsection within the Materials and Methods section of our manuscript.

The problems regarding the data and statistical analysis might still be caused by the poor quality of our images, which have caused inconvenience to you. As you noted, the charts are blurry with unreadable details, making independent evaluation impossible. We are very sorry for this, and have carefully revised all the existing problems this time and hope it meets your requirements and achieves the desired effect. We appreciate the reviewer’s insightful comments, which have greatly helped improve our manuscript.

Reviewer #2

Comment: Accept in present form

As the results suggests NLGN3 protein interacts with ziprasidone to form stable complexes, which may activate the NLGN3-NRXN signaling pathway in OPCs and enhance synaptic remodeling by reducing neuroinflammatory responses.

Response: Thank you for your valuable time and effort in reviewing our manuscript. We are delighted to learn that the manuscript is accepted in its present form. Thank you once again for your valuable comments and for recognizing the merit of our work.

Thank you again for your valuable comments and suggestions, and hope that we have fully addressed the questions and comments satisfactorily and the revised version can be accepted accordingly as soon as possible. We greatly appreciate the opportunity to revise our manuscript and sincerely hope that the revised version meets the requirements for publication in “PLOS One”.

Sincerely yours,

Dr. Guiqin Bai

Department of Basic Medicine and Forensic Medicine, North Sichuan Medical College, Nanchong 637000, Sichuan, China

E-mail: bgq123@nsmc.edu.cn

Tel: +86 18281731246

---

## [Decision Letter · Decision Letter 1]

6 Apr 2026

Single-cell multiomics data analysis of potential receptors and therapeutic drugs for epilepsy patients comorbid with depression

PONE-D-25-56676R1

Dear Dr. Bai,

We’re pleased to inform you that your manuscript has been judged scientifically suitable for publication and will be formally accepted for publication once it meets all outstanding technical requirements.

Kind regards,

Peng Zhong, Ph.D.

Academic Editor

PLOS One

Reviewers' comments:

Reviewer's Responses to Questions

**Comments to the Author**

Reviewer #2: All comments have been addressed

2. Is the manuscript technically sound, and do the data support the conclusions?

Reviewer #2: Yes

3. Has the statistical analysis been performed appropriately and rigorously?

Reviewer #2: Yes

4. Have the authors made all data underlying the findings in their manuscript fully available?

Reviewer #2: Yes

5. Is the manuscript presented in an intelligible fashion and written in standard English?

Reviewer #2: Yes

Reviewer #2: Accept in present form

As the results suggests NLGN3 protein interacts with ziprasidone to form stable

complexes, which may activate the NLGN3-NRXN signaling pathway in OPCs and

enhance synaptic remodeling by reducing neuroinflammatory responses

.

Reviewer #2: **Yes:** Dr. Saumya PatelDr. Saumya PatelDr. Saumya PatelDr. Saumya Patel

---

## [Editor Report · Acceptance letter]

PONE-D-25-56676R1

PLOS One

Dear Dr. Bai,

I'm pleased to inform you that your manuscript has been deemed suitable for publication in PLOS One. Congratulations! Your manuscript is now being handed over to our production team.

Kind regards,

on behalf of

Dr. Peng Zhong

Academic Editor

PLOS One